# Endothelial-to-Mesenchymal Transition in Atherosclerosis: Friend or Foe?

**DOI:** 10.3390/cells11192946

**Published:** 2022-09-21

**Authors:** Sarin Gole, Svyatoslav Tkachenko, Tarek Masannat, Richard A. Baylis, Olga A. Cherepanova

**Affiliations:** 1Cardiovascular and Metabolic Sciences, Lerner Research Institute, Cleveland Clinic, 9500 Euclid Avenue, NB5, Cleveland, OH 44195, USA; 2Genetics and Genome Sciences, Case Western Reserve University, 2109 Adelbert, RD, BRB, Cleveland, OH 44106, USA; 3Department of Medicine, Massachusetts General Hospital, 55 Fruit St Gray 730, Boston, MA 02114, USA

**Keywords:** endothelial-to-mesenchymal transition, atherosclerosis, plaque stability

## Abstract

Despite many decades of research, complications of atherosclerosis resulting from the rupture or erosion of unstable plaques remain the leading cause of death worldwide. Advances in cellular lineage tracing techniques have allowed researchers to begin investigating the role of individual cell types in the key processes regulating plaque stability, including maintenance of the fibrous cap, a protective collagen-rich structure that underlies the endothelium. This structure was previously thought to be entirely derived from smooth muscle cells (SMC), which migrated from the vessel wall. However, recent lineage tracing studies have identified endothelial cells (EC) as an essential component of this protective barrier through an endothelial-to-mesenchymal transition (EndoMT), a process that has previously been implicated in pulmonary, cardiac, and kidney fibrosis. Although the presence of EndoMT in atherosclerotic plaques has been shown by several laboratories using EC-lineage tracing mouse models, whether EndoMT is detrimental (i.e., worsening disease progression) or beneficial (i.e., an athero-protective response that prevents plaque instability) remains uncertain as there are data to support both possibilities, which will be further discussed in this review.

## 1. Introduction

EndoMT was initially observed during normal embryonic heart development, where endocardial EC were found adapting a mesenchymal-like phenotype with migratory properties to facilitate proper heart valve development [1,2]. EndoMT is broadly defined as a change in EC morphology, loss of protective eNOS, and activation of EndoMT markers such as mesenchymal markers (ZEB2, SLUG, VIM), SMC markers (ACTA2, TAGLN, CNN1), extracellular matrix and pro-inflammatory proteins, and enhanced proliferation and migration. In contrast to the epithelial-to-mesenchymal transition (EMT), a process that has been carefully explored given its role in development and cancer biology, the molecular mechanisms responsible for EndoMT are not well understood. Several studies have shown that the TGFβ/SMAD/SNAIL1 [3], IL1β/NFkβ [4], PDGFRβ/NFkβ/HIF1α [5], WNT/β-catenin [6], and ET1 [7] axes induce EndoMT in different in vitro and in vivo models (reviewed in the same issue of *Cells* by Peng et al. [8]). 

Advanced atherosclerotic plaques are composed of a highly thrombotic lipid-rich necrotic core covered by a highly cellular, collagen-rich structure called the fibrous cap [9]. Fibrous cap rupture (i.e., physical disruption of the cap) and plaque erosion (i.e., denuded endothelium with exposure of sub-endothelial collagen) are the two major pathophysiological mechanisms that cause the clinical thrombotic consequences of atherosclerosis, which include myocardial infarction and a subset of cerebrovascular accidents [10]. Until recently, ACTA2^+^ SMC were believed to be the only cells responsible for the formation of a stable, collagen-rich fibrous cap. However, several groups using EC-lineage tracing mouse models have demonstrated that during the development of atherosclerosis, EC within the fibrous cap region undergo EndoMT and become an additional source of ACTA2^+^ cells [11,12].

It is critical to note that recent EC- and SMC-lineage tracing studies combined with the single cell RNA-sequencing (scRNAseq) approach revealed many ambiguities in using traditional protein cell lineage markers for detecting EndoMT in atherosclerotic plaques. Thus, recent scRNAseq analysis on CD31^+^CD45^−^ cells (presumably EC) FACS-sorted from aortas of atherosclerotic *Apoe*^−/−^ mice was unable to distinguish between EC and SMC as phenotypically modified SMC within atherosclerotic lesions activate CD31 [13]. Moreover, using EC-lineage tracing mice, researchers found that only a fraction of EC-derived cells in pathological conditions, such as atherosclerosis and vein graft modeling, are CD31^+^ or CDH5^+^ [11,14]. Therefore, permanent EC-lineage tracing is essential to study EndoMT in pathological conditions.

At present, several EC-lineage tracing mouse models have been used to assess EndoMT in vivo, including inducible conditional models such as end. *Slc*-CreERT2 [11] and *Cdh5*-CreERT2 [15]. Figure 1 represents EndoMT EC within the fibrous cap area of the brachiocephalic artery lesions of EC-lineage tracing, *Cdh5*-CreERT2 Rosa-Stop-YFP *Apoe*^−/−^, mice fed a high-fat western diet (WD) for 18 weeks (advanced stage of atherosclerosis). Using these EC lineage tracing male mice, we found that an average of ~15% [3–23%] of ACTA2^+^ cells within the fibrous cap are of EC- and not SMC-origin, which account for ~13% [8–21%] of the total number of YFP^+^ (EC-derived) cells [16]. Intriguingly, we also observed that female mice had only 3% [0–9%] of ACTA2^+^YFP^+^ EC within the 30 µm fibrous cap area of the brachiocephalic artery from 18 weeks WD-fed mice [16], suggesting that these EC transitions may be partially responsible for sex differences in atherosclerosis development. Furthermore, recent scRNAseq analyses from several labs [13,17], including ours [16], confirmed that EC undergo EndoMT in atherosclerotic lesions. On the other hand, SMC-lineage tracing studies have been conducted in human coronary artery lesions based on the detection of the permanent epigenetic H3K4diMe mark on the *MYH11* promoter [18]. These studies demonstrated that a quarter of the ACTA2^+^ fibrous cap cells come from a non-SMC source, with the majority of CD31^+^ACTA2^+^ presumably EC [19]. Taken together, these studies strongly suggest that EndoMT accounts for ACTA2^+^ cells in both human and mouse atherosclerotic lesions. However, from the data discussed thus far, it remains unclear whether EndoMT is protective or detrimental. 

## 2. Potential Detrimental Role of EndoMT in Atherosclerosis

Hypotheses asserting a detrimental role for EndoMT arose from several studies. Thus, Dr. Michael Simons’ group found that EC-specific knockout of *Fgfr2* in atherosclerotic *Apoe*^−/−^ mice activated TGF-β signaling, leading to the extensive development of EndoMT and changes consistent with decreased plaque stability, including an increase in necrotic core size and a decrease in the size of the fibrous cap [12]. Furthermore, the same group found that the EC-specific knockout of *Tgfbr1/Tgfbr2* displayed reduced EndoMT in parallel with smaller atherosclerotic lesions and reduced vascular wall inflammation [13]. In addition, Dr. Jason Kovacic’s group used an EC-lineage tracing *Apoe*^−/−^ mouse model to demonstrate that a large fraction of EC acquires mesenchymal markers, Fibroblast Activation Protein (FAP) and Fibroblast Specific Protein 1 (FSP1), and that these cells are abundant within lesions [11]. Perhaps the most convincing evidence that EndoMT is deleterious was the correlation that both groups found between CD31^+^ACTA2^+^ cell prevalence (presumably EC undergoing EndoMT) and the severity of atherosclerosis in human atherosclerotic lesions [11,12]. However, these observational studies are unable to show whether EndoMT contributed to plaque progression or it is a compensatory mechanism actually relieving the already complicated atherosclerotic state. Taking their mouse and human data together, both groups concluded that their findings supported a detrimental role of EndoMT in atherosclerosis, although the distinction between correlation events and causality was not preformed.

Along these lines, Mehta et al. have discussed the potential role of EndoMT in atherosclerotic plaque initiation [20]. The authors described their discovery of the ALK5 mechanoreceptor, which responds to shear stress on vascular EC, and demonstrated that ALK5 could initiate EndoMT. ALK5 accomplishes this via the mediator protein SHC, a knockout of which displayed reduced EndoMT. However, the authors noted that the observed reduction in atherosclerosis within the *Alk5* knockout mice could not be concluded to be a direct result of lower EndoMT rates. In addition, we recently found that EC-specific knockout of the pluripotency factor OCT4 exacerbated atherosclerosis development in *Apoe*^−/−^ mice, including altered plaque characteristics consistent with decreased plaque stability [16]. We also demonstrated that the loss of OCT4 resulted in marked increases in EndoMT both in vivo and in vitro. Finally, Liang and co-authors demonstrated that Tenascin-X mediates disturbed flow-induced suppression of atherosclerosis and EndoMT by blocking TGFβ signaling [21]. In addition, several other studies that did not use an unbiased genetic EC-lineage tracing approach demonstrated that some compounds and clinical drugs that reduce atherosclerosis development, such as RGFP966 [22] (inhibitor of histone deacetylates 3), Icariin [23] (a compound derived from *Epimedium*), and simvastatin [24], potentially result in reduced EndoMT rates (reviewed by Huang et al. [25]). 

Although all these results show a clear correlation between the frequency of EndoMT and the severity of atherosclerosis, they do not answer whether these transitions are the cause or consequence of atherosclerosis. 

## 3. Potential Athero-Protective Role of EndoMT

While much evidence has been presented to demonstrate that EndoMT occurs in advanced atherosclerotic lesions, it is important to understand the clinically dangerous aspects of atherosclerotic pathogenesis to make conclusions about EndoMT’s role in cardiovascular health overall. The pathophysiological mechanisms of plaque rupture have been thoroughly studied. It is now well-accepted that plaque ruptures result from a thin, unstable fibrous cap, typically characterized by macrophage enrichment and depletion in ACTA2^+^ cells and mature collagen fibers. An investigation into causes of coronary deaths by Virmani et al. [26] found that ruptures of thin-fibrous cap atheroma were the cause of ~60% of acute thrombi in 200 sudden death cases. A classic study by Davies et al. [27] compared the cellular composition of ruptured plaques to intact plaques. The fibrous cap of ruptured plaques contained a smaller percentage of ACTA2^+^ cells (2.8% vs. 11.8%) and a large fraction of CD68^+^ cells (13.7% vs. 2.9%) when compared to intact atherosclerotic plaques. 

Assuming fibrous cap composition and stability are the most clinically significant characteristics in preventing catastrophic plaque rupture events, it is important to observe the evidence presented regarding EndoMT’s contribution to fibrous cap development. As described earlier, EndoMT has been implicated in multiple diseases, where it has been shown to contribute to pathogenic tissue fibrosis. However, unlike most pathogenic states, atherosclerosis relies on an intact fibrotic response to maintain a protective fibrous cap, and therefore disrupting processes contributing to this fibrosis could lead to plaque instability. Interestingly, a recent scRNAseq study on EC and SMC sorted from carefully dissected advanced atherosclerotic lesions from the EC- and SMC-lineage tracing *Apoe*^−/−^ mice revealed that EC give rise to cells transcriptomically similar to ACTA2^+^ SMC-derived cells; likely via EndoMT [17]. Going back to the idea that having higher rates of ACTA2^+^ cells in the fibrous cap is beneficial for lesion stability, these scRNAseq results would suggest that EndoMT may play an athero-protective role in the lesion. 

Regarding the origin of fibrous cap cells, recent data from our collaborative studies with Dr. Gary Owens’ laboratory found that ~15–20% of ACTA2^+^ cells in the fibrous cap of advanced brachiocephalic lesions were derived from EC that had undergone EndoMT [16,19], and that the number of ACTA2^+^ EC increased dramatically to ~60–70% following radiation-induced impairment of SMC investment [28], thus strengthening the case of EndoMT being a compensatory mechanism. These data were supported by genetic inactivation of *Pdgfbr* in SMC that led to the SMC-deficient brachiocephalic artery lesions and an observed ~40% increase in the EndoMT rates within the fibrous cap. Despite a substantial reduction in SMC content within the fibrous cap, plaque stability was maintained at early time points, suggesting that EC were capable of transiently compensating for the absence of SMC [19]. In addition, Dr. Owens’ group showed that the regression of advanced atherosclerotic lesions in response to long-term chow diet feeding is associated with increased ACTA2^+^ EndoMT EC [29]. Altogether, these results contradict the conclusions regarding the detrimental role of EndoMT in atherosclerosis.

## 4. Multiple Sub-Phenotypes of EndoMT in Atherosclerosis 

As stated earlier, researchers have used a broad range of mesenchymal, SMC, and pro-inflammatory markers to define EndoMT. As our understanding improves, broad phenotypic definitions are being replaced by a more complicated but precise classification. Vascular SMC are a prime example. For many years, SMC phenotypic switching/dedifferentiation has been considered a one-way transition (from contractile SMC to synthetic SMC). However, recent lineage tracing and scRNAseq studies have revealed that SMC-derived cells within mouse and human atherosclerotic arteries exhibit a far greater phenotypic diversity than previously has been believed. Using scRNAseq on mouse atherosclerotic lesions from SMC-lineage tracing atherosclerotic mice, several groups have identified at least seven unique SMC phenotypes, including potentially athero-protective SMC phenotypes (e.g., fibromyocytes, myofibroblast-like SMC) and athero-promoting phenotypes (e.g., osteoclast-like, macrophage-like SMC) [17,30,31,32]. From this precedent, it is possible that the multitude of transitions used to characterize EndoMT presents a much more complex set of phenotypes than previously believed. 

To test this hypothesis, we reanalyzed the publicly available scRNAseq dataset [30] on unsorted cells isolated from atherosclerotic human coronary arteries (Figure 2; Appendix A). Next, across the *CDH5*^+^*PECAM1*^+^ EC cluster, we performed an additional deeper cluster analysis, which identified at least two unique EndoMT clusters. Intriguingly, one EndoMT cluster was enriched with SMC contractile marker genes, including *ACTA2*, *TAGLN*, and *CCN1*, while the other ACTA2-negative EndoMT cluster was enriched with *FN1*, and extracellular matrix genes, including *COL8A1* and *BGN* (Figure 2b–d). Notably, both ACTA2 and fibronectin (FN1) are considered potential markers of EndoMT in atherosclerosis. However, their functions are potentially very different in that fibronectin (FN1) promotes EC activation and early plaque formation via NFkB-dependent pro-inflammatory mechanisms [33,34,35], andACTA2 is responsible for the contractile phenotype. Therefore, we hypothesize that EndoMT is not a single transition. Similar to SMC phenotypic switching, EC can shift toward different phenotypic states in pathological conditions, including ACTA2^+^ (athero-protective in the fibrous cap) or ACTA2^−^ (e.g., migratory/angiogenic/pro-inflammatory) states.

Simons et al. have shown that reduced EndoMT marker expression due to EC-specific *Tgfbr1/Tgfbr2* knockout resulted in decreased occlusion and regression of plaques [13]. These assertions might be erroneous in the sense that Simon et al.’s scRNAseq profile chart (see *Figure 4 in ref*. [13]) of the differentially expressed genes displayed marked decreases in pro-inflammatory EndoMT markers (*Fn1*, *Serpine1*, *Fbln5*), MMPs, and collagens (*Col8a1*, *Col3a1*, etc.), but showed no obvious differences in SMC contractile genes. Moreover, the loss of *Tgfbr1/Tgfbr2* in EC led to observable increases in the expression of SMC contractile genes; *Acta2* and *Cnn1*. In these observations, we can see how EndoMT, in its current conception, could be too broad of a characterization, which might help us understand how we need to go about evaluating its role in atherosclerosis in the future.

## 5. Conclusions

EndoMT is an important source of ACTA2^+^ cells within the atherosclerotic lesion fibrous cap. However, whether EndoMT is an athero-protective or deleterious process remains uncertain. Given that there is convincing evidence to support both directions, it is almost certain that the answer is “it depends.” Furthermore, it is clear that there are discrepancies in how EndoMT is characterized, which offer no help in discerning EndoMT’s role and potential as a therapeutic target in the future. In prior literature, different research groups have proposed the potentially protective or detrimental roles of EC expressing mesenchymal, SMC, and pro-inflammatory markers in the initiation of atherosclerosis pathogenesis. However, critical challenges for future studies will be to elucidate the role of different subpopulations of EndoMT EC (e.g., ACTA2^+^ versus ACTA2^−^) and identify environmental cues that regulate these EC phenotypic transitions at the different stages of atherosclerosis development. Additionally, a study temporally discerning worsening of atherosclerosis and development of different types of EndoMT would be extremely valuable. Another question remains to be investigated—whether the EndoMT EC-enriched fibrous cap areas demonstrate local mechanical properties consistent with the increased mechanical stability that is observed in fibrous cap areas enriched with the athero-protective ACTA2^+^ SMC.

Studies of EndoMT in mouse models of atherosclerosis have demonstrated that EC-lineage tracing is critical to identify EndoMT in vivo. Therefore, the development of new technologies for unbiased identification of EndoMT EC in human atherosclerotic plaques is of great importance for developing EndoMT-dependent therapeutics in humans. More studies are also needed to determine how EndoMT might be therapeutically manipulated to increase plaque stability without inducing detrimental effects in other cell types. It is important to understand that atherosclerosis is more likely to be intervened with rather than prevented in a clinical setting; therefore, it might be beneficial to further explore the potential athero-protective role of EndoMT. If EndoMT, or its subtype, is shown to unambiguously increase plaque stability, we obtain a great therapeutic target that could be important even for late stages of atherosclerosis.

## Figures and Tables

**Figure 1 cells-11-02946-f001:**
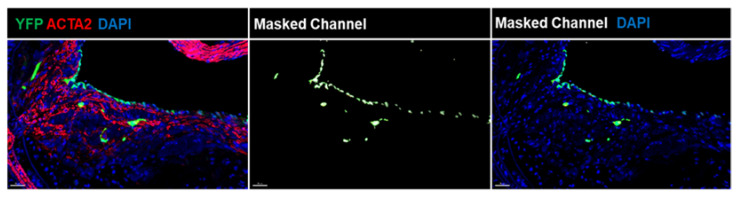
EndoMT in mouse atherosclerotic lesions. Immunofluorescence (IF) confocal z-stacks were analyzed using Microscopy Image Analysis Software (IMARIS) to demonstrate the unbiased co-localization for YFP and ACTA2 pixels. Ten-micrometer brachiocephalic artery cross-sections from EC-lineage tracing (*Cdh5*-CreERT2 Rosa-Stop-YFP *Apoe*^−/−^ mice received 10 tamoxifen injections between 6–8 weeks of age and fed a high-fat WD for 18 weeks were stained for YFP (EC lineage tracing), ACTA2, and DAPI). Representative IF z-stacks were analyzed in IMARIS to visualize the YFP/ACTA2 pixel co-localization. Based on this co-localization, the masked channel was made. Next, the masked channel was applied to the DAPI channel to check for the single cell location. Scale bar = 30 µm.

**Figure 2 cells-11-02946-f002:**
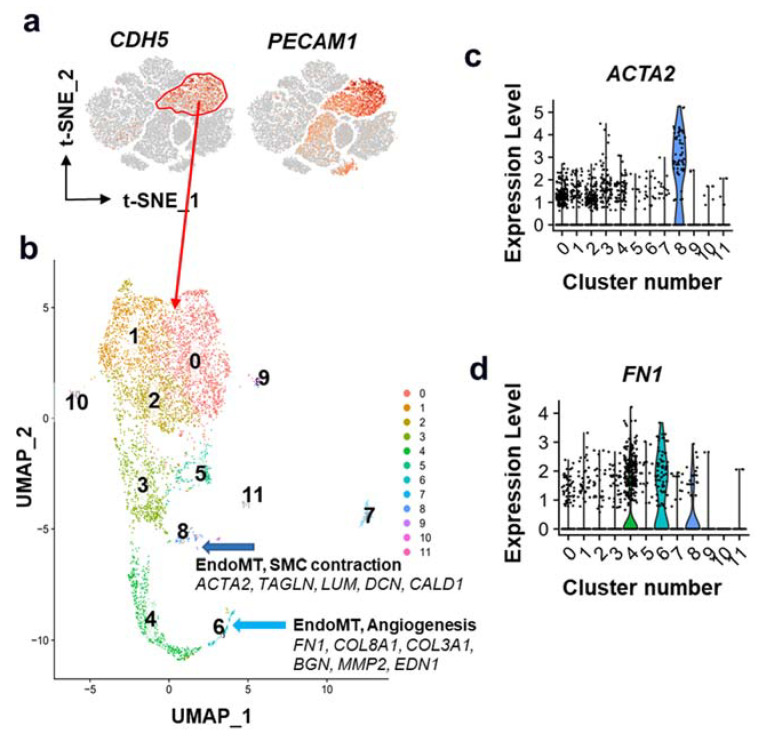
EndoMT in human coronary artery atherosclerosis. Publicly available scRNAseq data from Wirka et al. [30] on unsorted cells from atherosclerotic human coronary arteries were reanalyzed. (**a**) EC cluster was determined based on the expression of *CDH5* and *PECAM1* (red oval). (**b**) Cells from the EC cluster were further analyzed using higher cluster resolution. We observed two clusters, 6 and 8, enriched with EndoMT markers. (**c**) Interestingly, only cluster 8 had *ACTA2*^+^ and other SMC contractile marker-positive cells. (**d**) Cluster 6 was enriched with *FN1*, collagens, and pro-angiogenesis genes.

## Data Availability

Not applicable.

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
