# Peer review of "Endothelial-to-Mesenchymal Transition in Atherosclerosis: Friend or Foe?"

_cells, 2022, doi:10.3390/cells11192946_

Round 1

Reviewer 1 Report

This manuscript reviews the current knowledge on endothelial-to-mesenchymal transition (EndoMT) and discusses its potential roles in atherosclerosis. This review comes out in a timely manner and I believe it should attract many readers of the journal. There are several minor issues to be addressed here.

1. As the authors mentioned in the “Introduction”, molecular mechanisms of EndoMT is less understood compared to EMT. It will be helpful if the authors can talk about known factors that induce EndoMT during atherogenesis and, ideally, explain the extracellular signals and the intracellular signaling pathways involved.

2. Figure 2 shows the scRNA-seq data re-analyzed by the authors based on another reference. As the details of how scRNA-seq data are analyzed often significantly affect the end results, it is highly recommended that the authors write in a supplemental file to describe how they re-analyzed the data and generate Figure 2. For example, how many human artery samples were used? How sequencing data were aligned and checked for quality? What software was used?

3. Are there protein markers for the EndoMT in vivo or any assays available for detecting EndoMT in vivo?

4. It would be helpful to list the open questions to be answered in the field of EndoMT in atherosclerosis so that readers get a sense of the future directions at the end of the paper.

5. The first section “1. Introduction” is followed by “3. Potential athero…”. So the section needs to be renumbered.

Reviewer 2 Report

Authors presented an interesting review of the current literature on the topic of atherosclerosis and the role of Endothelial-to-Mesenchymal Transition (EndoMT) in the formation of fibrous cap.

The review is well written and quite interesting. I do not have major concerns. However, I suspect there is a mistake in the numbering of the section: after #1 next section is #3.

Author Response

We greatly appreciate the Reviewer's thoughtful Review and high enthusiasm about our manuscript. In addition, we apologize for the mistake with the missing Title for section 2. Section 2 Title, "Potential detrimental role of EndoMT in atherosclerosis", was added to the revised version of the manuscript.

Reviewer 3 Report

 Gole et al. reviewed the current knowledge about endothelial to mesenchymal cellular events in the atherosclerotic disease pathogenesis entitled Endothelial-to-Mesenchymal Transition in Atherosclerosis: 2 Friend or Foe? This review paper is undoubtedly beneficial to understand the fibrosis event, a severe consequence of compromise of organ function. For example, cardiac fibrosis-derived stiffness eventually causes cardiac dysfunction. I have few concerns that need to address by the authors:

1) why EndoMT is not an exclusively detrimental event in disease biology but instead is considered as a defensive mechanism

2) Authors should discuss how the EndoMT cellular event contributes to inflammation 

3) Authors need some expansion in the discussion part to address how cardiac fibrosis and EndoMT are linked at the angle of atherosclerotic disease pathogenesis. The origin of a-SMA should also be focused on in the discussion part through consideration of the EndoMT event.  
